# Contrast Enhancement in MRI Using Combined Double Action Contrast Agents and Image Post-Processing in the Breast Cancer Model

**DOI:** 10.3390/ma16083096

**Published:** 2023-04-14

**Authors:** David MacDonald, Frank C. J. M. van Veggel, Boguslaw Tomanek, Barbara Blasiak

**Affiliations:** 1Institute of Nuclear Physics Polish Academy of Science, Radzikowskiego 152, 31-342 Krakow, Poland; david.macdonald@ifj.edu.pl (D.M.); tomanek@ualberta.ca (B.T.); 2Department of Chemistry, Centre for Advanced Materials & Related Technologies (CAMTEC), University of Victoria, Victoria, BC V8P 5C2, Canada; fvv@uvic.ca; 3Division of Medical Physics, Department of Oncology, University of Alberta, 8303 112 St NW, Edmonton, AB T6G 2T4, Canada; 4Department of Clinical Neurosciences and Radiology, University of Calgary, 2500 University Drive NW, Calgary, AB T2N 1N4, Canada

**Keywords:** MRI, contrast agents, imaging

## Abstract

Gd- and Fe-based contrast agents reduce T_1_ and T_2_ relaxation times, respectively, are frequently used in MRI, providing improved cancer detection. Recently, contrast agents changing both T_1_/T_2_ times, based on core/shell nanoparticles, have been introduced. Although advantages of the T_1_/T_2_ agents were shown, MR image contrast of cancerous versus normal adjacent tissue induced by these agents has not yet been analyzed in detail as authors considered changes in cancer MR signal or signal-to-noise ratio after contrast injection rather than changes in signal differences between cancer and normal adjacent tissue. Furthermore, the potential advantages of T_1_/T_2_ contrast agents using image manipulation such as subtraction or addition have not been yet discussed in detail. Therefore, we performed theoretical calculations of MR signal in a tumor model using T_1_-weighted, T_2_-weighted, and combined images for T_1_-, T_2_-, and T_1_/T_2_-targeted contrast agents. The results from the tumor model are followed by in vivo experiments using core/shell NaDyF_4_/NaGdF_4_ nanoparticles as T_1_/T_2_ non-targeted contrast agent in the animal model of triple negative breast cancer. The results show that subtraction of T_2_-weighted from T_1_-weighted MR images provides additional increase in the tumor contrast: over two-fold in the tumor model and 12% in the in vivo experiment.

## 1. Introduction

Magnetic resonance imaging (MRI) provides the best soft tissue contrast among diagnostic imaging modalities such as computed tomography (CT), positron emission tomography (PET), or X-ray. The contrast provided by MRI is based on proton density as well as the T_1_ and T_2_ relaxation times of an imaged object. MRI technique utilizes these parameters for contrast manipulation by applying proton density, T_1_-weighted (T_1_w) or T_2_-weighted (T_2_w) MR images. MRI contrast may be provided solely by tissues themselves, due to differences in their relaxation times. Contrast agents shortening T_1_ and/or T_2_ relaxation times improve the detection of small pathologies such as those seen in early stages of cancer. The most commonly used T_2_w MRI pulse sequences are based on spin echo (SE) or gradient echo (GE) with long repetition time (TR) and long echo time (TE) [1]. For fast T_1_w, inversion recovery (IR) pulse can be added before a pulse sequence (e.g., IR-GE) [2].

Tissues with short T_1_ relaxation provide positive (hyperintense) contrast, while tissues with short T_2_ relaxation provide a lower signal hence negative (hypointense) contrast. Contrast agents, shortening T_1_ and/or T_2_, accumulating in tissues of interest, such as tumors, are used to enhance these effects to improve the diagnostic capabilities of MRI, especially for small biological targets [3].

Due to high r_1_ relaxivity (r_1_ = 1/T_1_) Gd^3+^-based particles are commonly used as T_1_ contrast agents [4] while superparamagnetic iron oxide nanoparticles (SPION) are used as T_2_ contrast agents due to their high r_2_ relaxivity (r_2_ = 1/T_2_) [5]. The large magnetic moments of T_2_ contrast agents cause high signal change per unit of particles, therefore small quantities of SPION are sufficient to obtain strong contrast imaging [6,7,8]. Furthermore, SPIONS are less toxic than Gd^3+^. Their surface can be functionalized enhancing their specificity, biocompatibility, and biodegradability [6,9,10]. SPION can be employed for targeted drug delivery to cancer cells and other disease sites by synthesizing them with target-specific proteins [6,7,10]. While SPION can be relatively easily functionalized, they reduce mostly T_2_ relaxations, providing negative contrast which limits their diagnostic capability [5].

Recently, core/shell nanoparticles have been produced and used as contrast agents due to their capability of reducing both T_1_ and T_2_ relaxation times. The nanoparticles that reduce both T_1_ and T_2_ relaxation times have been used as contrast agents to improve diagnosis of various diseases, for example: atherosclerotic plaque in the carotid arteries [11], liver metastases [12], identification of white matter lesions in multiple sclerosis (MS) [13], prostate cancer in animal models [14], or in functional MRI to improve the contrast between brain tissues [15]. The core of the dual contrast agents consisting of inorganic material can be covered with either inorganic materials (e.g., silica or gold) or organic materials (phospholipids, fatty acids, polysaccharides, peptides, or other surfactants and polymers) [3,16]. The advantages of some of these nanoparticles include lower toxicity, greater dispersibility, biocompatibility, easier conjugation with other bioactive molecules, and increased thermal and chemical stability [3,17]. These properties can be controlled by the selection of their composition and size. For example, iron oxide was doped with Gd^3+^, and NaDyF_4_ nanoparticles (NPs) were covered with NaGdF_4_ showing high r_1_ and r_2_ relaxivity [18]. In the NaDyF_4_/NaGdF_4_ contrast agent, the dysprosium (Dy^3+^) ion provides T_2_ contrast while Gd^3+^ provides mostly T_1_ contrast [19]. Hybrids of nanocubes with both iron and gadolinium as T_1_/T_2_ dual contrast agents were also produced and used in in vivo MRI [20]. It was also shown that the relaxivities of nanoparticles depend on their size [21].

While manipulation of the contrast agent’s composition and its size enables higher relaxivity, the maximum concentration hence toxicity remains to be the major obstacle in improving MRI diagnostic capabilities. To deal with this issue, Girard et al. proposed to apply a combination of MRI pulse sequences to utilize changes in both T_1_ and T_2_ relaxation times due to the presence of dual action (T_1_ and T_2_) contrast agents [22]. The authors used a spin echo (SE) and ultrashort echo (UTE) pulse sequences and calculated optimal pulse sequence parameters (echo time, repetition time, flip angle) for both sequences to obtain improved contrast [22].

So far, reports on contrast agents analyzed solely the changes in tumor signal after contrast agent injection, neglecting changes in the surrounding tissue [23]. However, from the diagnostic perspective, this approach is not optimal, as the tumor signal changes should be compared to signal changes in the adjacent tissues. Recently some authors proposed image post-processing to enhance the signal-to-noise ratio (SNR) of a tumor, defining SNR as the average tumor signal divided by the standard deviation of the noise [23]. While this approach is advantageous, it does not consider signal changes in both tumor and normal adjacent tissue, as contrast agents accumulate in both tumor and normal adjacent tissue. Therefore, we studied and compared changes in signals of both the tumor and normal adjacent tissue rather than the tumor SNR alone. We also performed a simple image manipulation of a phantom tumor model subtracting and adding T_1_w, and T_2_w images when T_1_/T_2_ “perfect” (i.e., accumulating in tumor only) targeted contrast agents were used. The results show that subtraction of T_1_w and T_2_w pre- and post-contrast MR images improves the tumor contrast. Theoretical calculations are followed by an example of in vivo MRI in a mouse model of triple negative breast cancer (TNB) at 9.4T using core/shell NaDyF_4_/NaGdF_4_ (21.2 nm/0.6 nm) contrast agent.

## 2. Materials and Methods

### 2.1. Theoretical Analysis of a Phantom Tumor Model

For theoretical analysis of tumor contrast enhancement in MRI accomplished with contrast agents, we considered the signal changes in T_1_w and T_2_w MR images caused by the application of T_1_, T_2_, and T_1_/T_2_ contrast agents. We assumed that the tumor is located within the normal adjacent tissue and that both the tumor and normal adjacent tissue have an elliptical shape and are homogenous (Figure 1). We defined tumor contrast (TC) as a ratio of signal from the tumor divided by the signal from adjacent tissue: TC = S_tu_/S_ti_, where S_tu_ is the average signal from the tumor, and S_ti_ is the average signal coming from the adjacent tissue.

Based on the previous reports on cancer MRI and, in particular, on MRI of breast cancer studies (e.g., [24]) showing about 50% TC enhancement caused by contrast agent injection, we considered a “perfect” tumor phantom model. This allowed us to study TC without influencing factors occurring in vivo (e.g., variable contrast absorption, tumor metabolism, tumor stage, contrast agent concentration etc.,) and MRI pulse sequence type and its parameters. We first assumed that tumor and surrounding tissue signals are identical in both T_1_w (equal to 100 arbitrary units (au)) and in T_2_w images (equal to 70 au). We also assumed “perfect” targeted contrast agents, accumulated in the tumor only. We further assumed, for T_1_ contrast agent, the signal increases by 50% in tumor only and does not change in the normal adjacent tissue. Similarly, for T_2_ contrast agent: signal from tumor decreases by 50% and is unchanged in normal adjacent issue in T_2_w MR images. For T_1_/T_2_ contrast agent, we assumed 50% signal increase and 50% reduction in T_1_w and T_2_w images, respectively [25,26]. In addition to providing signal values obtained from T_1_w and T_2_w images, we also added and subtracted images (Table 1).

### 2.2. Animal Model of Breast Cancer

For the mouse model of TNB, we used MDA-MB-231 cells [14]. Cells were cultured in RPMI supplemented with 5% fetal bovine serum (RPMI/5%) and maintained in a humidified 5% CO_2_ atmosphere at 37 °C. Cells were harvested by trypsinization (EDTA/trypsin), resuspended in RPMI/5%, and counted (BioRad TC10 counter, Hercules, CA, USA). About 2 × 10^6^ cells/mL of RPMI/5% were prepared and kept at 37 °C in a water bath up to the orthotopic implantation. At that point, cells were centrifuged (1500 rpm, 5 min) and resuspended in 20 μL sterile saline, and injected into the left #4 inguinal mammary gland of a 6-week old nu/nu female mouse. Implanted in the mouse mammary gland, MDA-MB-231 cells grew as solid spheroidal tumors over 3 weeks. Once the tumor had reached a size of ~50 mm^3^, the mouse (weight 35.6 g, heart rate 310–840 beats/min, body temp 36.5–38.0°) was injected via the tail vein with the contrast agent.

### 2.3. Synthesis and Characterisation of the NaDyF_4_/NaGdF_4_ Nanoparticles

The details of the NaDyF_4_/NaGdF_4_ synthesis are described elsewhere [14]. Briefly, prior to the synthesis of the core/shell NPs, cubic-phase (α) NaGdF_4_ NPs were synthesized, which acted as the sacrificial NPs for shell formation [27]. The multi-step process of the cubic (α) phase of sacrificial NaGdF_4_ synthesis started from gadolinium oxide, via gadolinium trifluoroacetate, mixed with various chemicals, heating, washing, and cooling cycles. Finally, the NP precipitated and washed with ethanol and dispersed in hexane. To synthesize NaDyF_4_, dysprosium (III) chloride hexahydrate was stirred in oleic acid and 1-octadecene under vacuum for 45 min at 120 °C. Further steps included heating, cooling, and stirring solutions of methanol containing sodium hydroxide and ammonium fluoride. After removing methanol by heating, the previously synthesized sacrificial (α) NaGdF_4_ nanoparticles dispersed in 1-octadecene (1 mL) were injected into the solution and stirred for 15 min to form a core-shell nanostructure. The nanoparticles were precipitated and washed with ethanol, and then finally dispersed in hexanes [14].

The synthesized NaDyF_4_/NaGdF_4_ NPs were analyzed using the X-ray diffraction method (XRD), showing the standard pattern of the hexagonal phase (β) of the NaDyF_4_ core. Analysis of the particle size distribution using the transmission electron microscopy (TEM) images showed a NaGdF_4_ shell thickness of 0.6 ± 0.1 nm while the core NaDyF_4_ NP showed a diameter of 21.2 ± 0.1 nm and the core/shell NPs had a diameter of 21.8 ± 0.1 nm. The relaxivities r_1_ and r_2_ of the NaDyF_4_/NaGdF_4_ NPs in deionized water were r_1_ = (9.0 ± 0.2) × 10^5^ mM^−1^ s^−1^ and r_2_ = (147.0 ± 7.5) × 10^5^ mM^−1^ s^−1^ respectively [14].

### 2.4. In Vivo MRI Experiments

A 9.4 T, 21 cm magnet bore MRI system (Bruker, Ettlingen, Germany) was used for the in vivo experiments. A volume radio frequency coil was placed over the animal covering the region of interest (ROI). The animal was imaged before and 15 min after the injection of NaDyF_4_/NaGdF_4_ (~21.2 nm/0.6 nm) non-targeted contrast agent. A total of 0.250 mL of the contrast agent was administered via the tail vein (vehicle, 0.9% saline) [14]. For the MRI, anesthesia was induced with 4% isoflurane and maintained with 2.7% isoflurane in 69% N_2_O and 30% O_2_ using a vaporizer. The animal experiment was approved by the local Animal Care Committee (number AC13-0202).

### 2.5. MR Imaging Parameters

To collect T_1_w and T_2_w in vivo MRI, we used rapid acquisition with relaxation enhancement (RARE) and multi-echo spin-echo (MESE) pulse sequences, respectively. The sequences were triggered with respiration to reduce possible motion artifacts. Parameters used for the T_1_w RARE pulse sequence were: echo time (TE) 7 ms, repetition time (TR) 750 ms, field of view (FOV) 2.56 × 2.56 cm, slice thickness 1 mm, and matrix 128 × 128. Parameters of the T_2_ weighted SE pulse sequence were: TE = 35 ms, TR = 5000 ms, FOV = 2.56 × 2.56 cm, slice thickness 1 mm, and matrix size 128 × 128. The images were collected before and 15 min post injection of the NaDyF_4_/NaGdF_4_ (average diameter of 21.2 nm and an average shell thickness of 0.6 nm) core/shell non-targeted contrast agent [14].

### 2.6. Selection of ROIs

The ROIs for the tumor and adjacent tissue were selected manually and an example of this is shown in Figure 2. The green line delineates the adjacent tissue and the red line delineates the tumor. Signal values S_tu_ and S_ti_ were calculated as the average of the signal intensities within ROI. Tumor contrast (TC) was analyzed in T_1_w and T_2_w MR images, as well as in subtracted and added T_1_w and T_2_w images before and after contrast agent injection. The subtraction and addition of images were performed using ImageJ (Bethesda, MD, USA) version 1.53 software in 16-bit format. The images were then converted to an 8-bit image jpeg format (0–255) and the analysis was completed with Python (Centrum Wiskunde & Informatica, Amsterdam, The Netherlands) version 3.6.

## 3. Results

### 3.1. Calculations of Contrast in a Tumor Phantom Model

The signals from the tumor and adjacent tissue as well as TC values before and after application of contrast agents are shown in Table 1. The corresponding images are shown in Figure 3, Figure 4, Figure 5 and Figure 6.

As seen in Table 1 and Figure 3, TC is the same and equals 1.00 before the contrast agent injection in T_1_w and T_2_w images as well as the subtracted and added images.

The TC values change in T_1_w, T_2_w, T_1_w − T_2_w, and T_1_w + T_2_w after injection of all contrast agents. For the T_1_ contrast agent, TC values are 1.50, 1.00, 2.67, and 0.71 in T_1_w, T_2_w, T_1_w − T_2_w, and T_1_w + T_2_w images, respectively. Figure 4A–D show the corresponding images. The highest TC = 2.67 is visible in the subtracted images (Figure 4C). The TC values change after injection of the T_2_ contrast agent and are 1.00, 0.47, 2.17, and 1.07 in T_1_w, T_2_w, T_1_w − T_2_w, and T_1_w + T_2_w images, respectively. Figure 5A–D show the corresponding images. The highest TC = 2.17 value is visible in the subtracted images (Figure 5C). TC values change after injection of the T_1_/T_2_ contrast agent and are 1.50, 0.50, 3.83, and 1.09 in T_1_w, T_2_w, T_1_w − T_2_w, and T_1_w + T_2_w images, respectively. Figure 6A–D show the corresponding images. The highest TC = 3.83 value is visible in the subtracted images (Figure 6C).

### 3.2. In Vivo Experiments

Figure 7 and Figure 8 show the MR images of a mouse bearing TNB tumor pre-injection (Figure 7) and post-injection (Figure 8) of the NaDyF_4_/NaGdF_4_ non-targeted contrast agent. Images marked with a, b, c, and d correspond to T_1_w, T_2_w, subtracted T_1_w − T_2_w, and added T_1_w + T_2_w, respectively. The signal and TC values in each case are shown in Table 2.

The results of in vivo experiments pre-injection (Figure 7) of the contrast agent show the highest TC value for subtracted T_1_w − T_2_w images (TC = 1.52) when compared to T_1_w, T_2_w and T_1_w + T_2_w images with TC equal to 1.15, 0.33, and 1.00, respectively. Injection of the non-targeted NaDyF_4_/NaGdF_4_ contrast agent into a mouse bearing TNB cancer increased the tumor contrast in the T_1_w image (Figure 8A), and decreased in the T_2_w image (Figure 8B), as expected. Image of the added T_1_w + T_2_w (Figure 8D) showed lower tumor contrast (1.09) compared to the T_1_w image (1.24) and higher contrast compared to that in the T_2_w image (0.32). The subtracted T_1_w − T_2_w image (Figure 8C) showed that the TC was the highest (1.56).

## 4. Discussion

Increasing tumor contrast is an ever-evolving pursuit for diagnostic imaging, specifically in MRI. Increasing the tumor contrast can lead to earlier tumor detection. While MRI techniques such as T_1_w and T_2_w imaging enable visualization of tumor regions, they are not sufficient for the detection of small tumors, and tumors with similar signals as surrounding tissues. To overcome this limitation T_1_, T_2_, and T_1_/T_2_ contrast agents have been applied. Of particular interest are T_1_/T_2_ contrast agents, as their surface can be used for conjugation with various delivery vehicles, such as antibody [28,29,30]. However, T_1_/T_2_ contrast agents provide lower TC in T_1_w images compared to T_1_-only contrast agents as their T_1_ shortening effect is usually smaller than that of the “pure“ T_1_ contrast. Similarly, shortening T_2_ is smaller for T_1_/T_2_ contrast agents than for “perfect” T_2_ contrasts, which reduce T_2_ only. These effects diminish the potential capabilities of the T_1_/T_2_ contrast agents.

The result for the phantom model showed that the application of the T_1_/T_2_ contrast agent alone improved tumor contrast by 50% (from 1.0 to 1.5) on T_1_-weighted images and reduced contrast by 50% (from 1.0 to 0.5) on T_2_w images. The image subtraction additionally improved contrast over two-fold (from 1.50 to 3.83) when compared to T_1_-weigthed. For the in vivo experiments, the corresponding values were: increase by 8% (from 1.15 to 1.24) in T_1_-weighted images, decrease by 1% (from 0.33 to 0.32) in T_2_-weighted images, and additional improvement due to image subtraction by 12% (from 1.24 to 1.56). Overall, the results showed that the image contrast enhancement caused by the T_1_/T_2_ contrast agents can be additionally improved by the application of image subtraction. Furthermore, TC is the highest for the T_1_/T_2_ contrast agent. Subsequently, potential drawbacks associated with lower contrast provided by the T_1_/T_2_ contrast agents can be rectified.

The differences between the phantom tumor model and in vivo TC results are due to the fact that our phantom tumor model assumes the application of the “perfect” targeted contrast agents, and our in vivo experiments use a non-targeted contrast agent. Application of the non-targeted contrast agent causes similar changes in signal in both tumor and normal adjacent tissue, while the phantom model shows signal changes in tumor only.

In the presented studies only one type of NPs was used to show the general concept of image analysis. The NaDyF_4_/NaGdF_4_ NPs used in the in vivo study had core/shell size of 21.2 nm/0.6 nm. However, we expect the observed increase in TC would be achieved independently of the size of the NP. NPs with higher r1 and r2 would provide even more evident increase in TC. The increase in r_1_ and r_2_ can be achieved by increasing both the shell and core sizes. In the case of the NaDyF_4_/NaGdF_4_, increase in the Gd shell size would increase r1, hence providing desired improvement in TC. The independence of the TC enhancement due to the applied image post-processing is also supported by the theoretical analysis of the phantom model, where an arbitrary contrast was considered. However, further studies with other NPs are required to find their optimum parameters (e.g., core/shell ratio and size) to obtain maximum TC.

## 5. Conclusions

The results show that the T_1_/T_2_ contrast agents can indeed provide improved diagnostic capability over the T_1_ and T_2_ contrast agents alone. However, to fully benefit from its application, the contrast should provide as strong as possible T_1_ and T_2_ shortening (maximal r_1_ and r_2_), should be targeted, and then T_1_w and T_2_w images should be collected and subtracted. These improvements hold the potential to enhance the diagnostic capabilities of contrast enhanced MRI.

## Figures and Tables

**Figure 1 materials-16-03096-f001:**
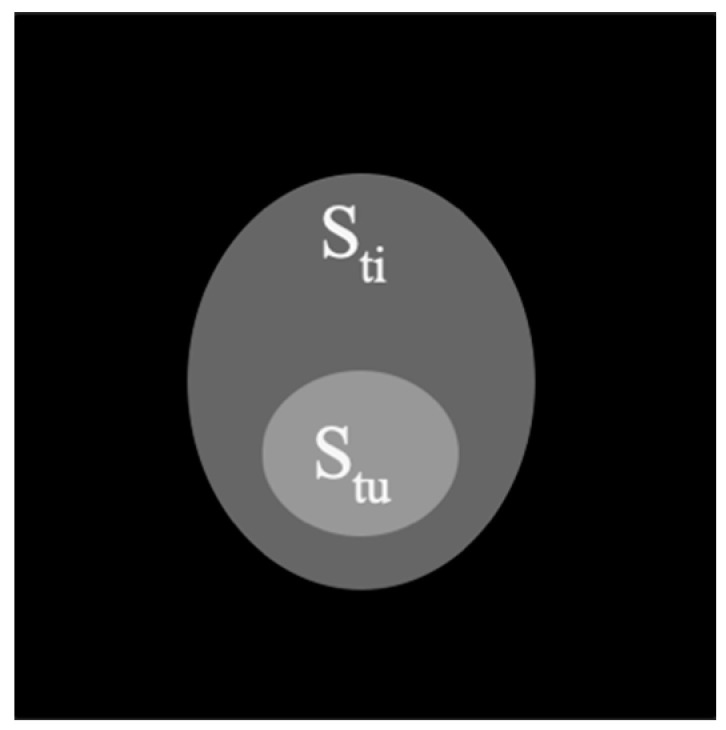
An image of a phantom tumor model comprising tumor (tu) surrounded by a normal adjacent tissue (ti) used for contrast analysis.

**Figure 2 materials-16-03096-f002:**
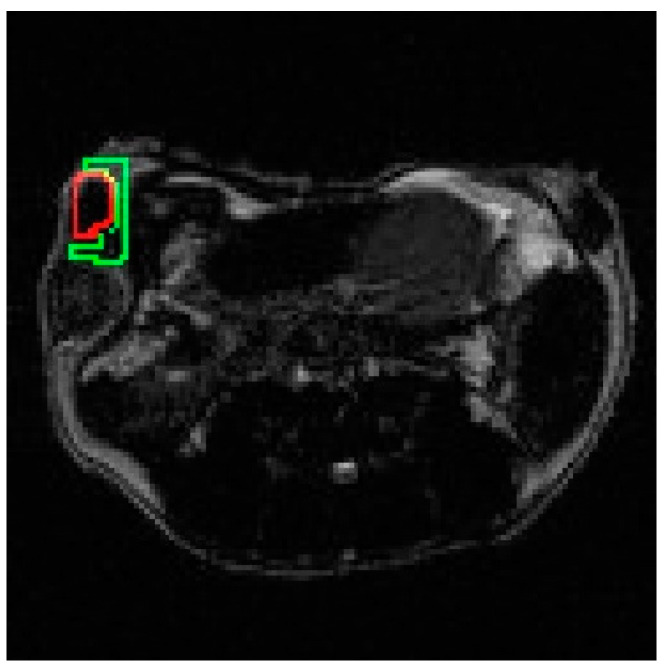
Selection of the ROIs in the mouse abdomen MRI: green line delineates the adjacent tissue, and the red line delineates the tumor.

**Figure 3 materials-16-03096-f003:**
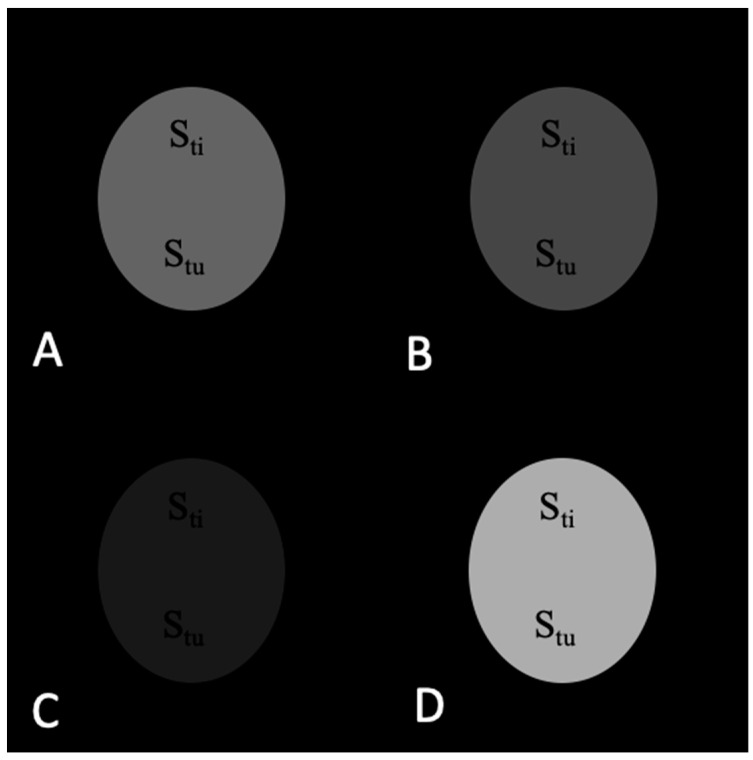
MR images of the tumor phantom model before contrast injection: (**A**) T_1_w. (**B**) T_2_w. (**C**) Subtracted T_1_w − T_2_w. (**D**) Added T_1_w + T_2_w. TC = 1.00 in all images.

**Figure 4 materials-16-03096-f004:**
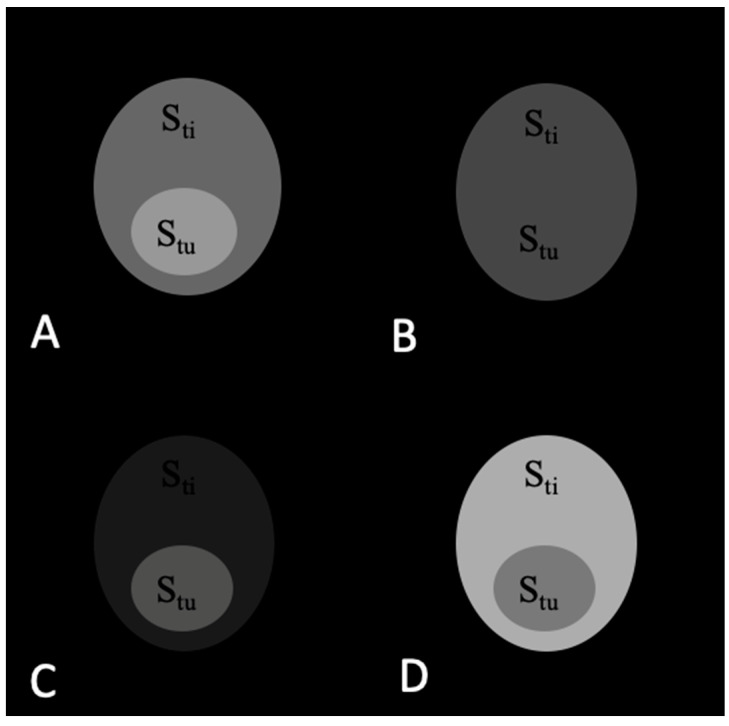
MR images of the tumor phantom model after injection of the T_1_ contrast agent. (**A**) T_1_w. (**B**) T_2_w. (**C**) Subtracted T_1_w − T_2_w. (**D**) Added T_1_w + T_2_w. The highest TC value is 2.67 in image (**C**).

**Figure 5 materials-16-03096-f005:**
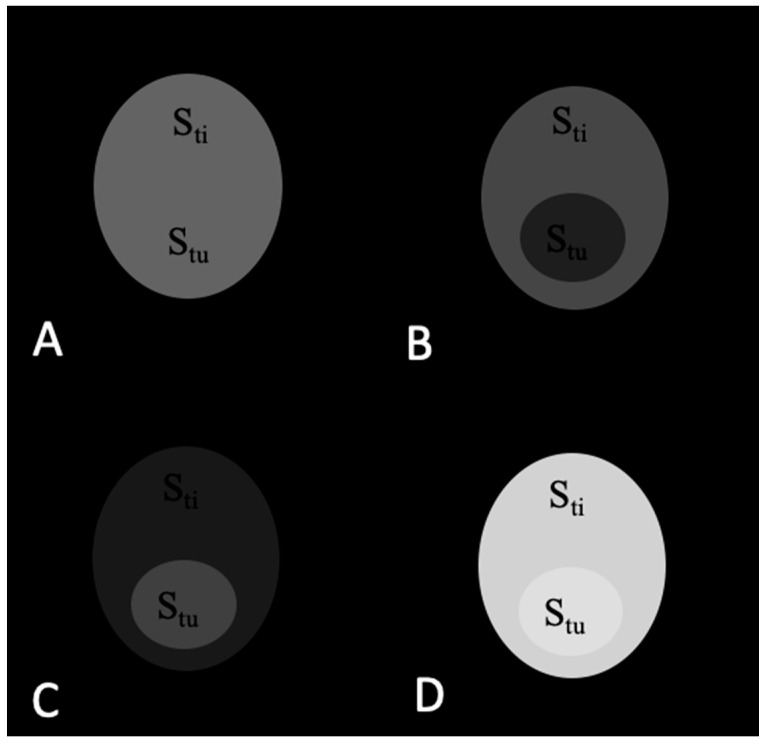
MR images of the tumor phantom model after injection of the T_2_ contrast agent. (**A**) T_1_w. (**B**) T_2_w. (**C**) subtracted T_1_w − T_2_w. (**D**) Added T_1_w + T_2_w. The highest TC value is 2.17 in image (**C**).

**Figure 6 materials-16-03096-f006:**
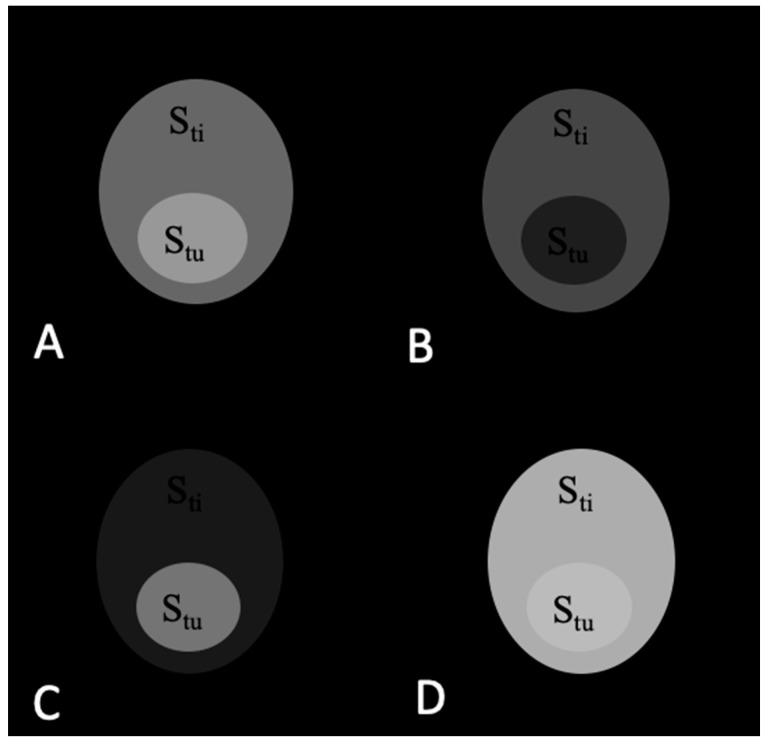
MR images of the tumor phantom model after injection of the T_1_/T_2_ contrast agent: (**A**) T_1_w. (**B**) T_2_w. (**C**) Subtracted T_1_w- T_2_w. (**D**) Added T_1_w+ T_2_w. The highest TC value is 3.83 in image (**C**).

**Figure 7 materials-16-03096-f007:**
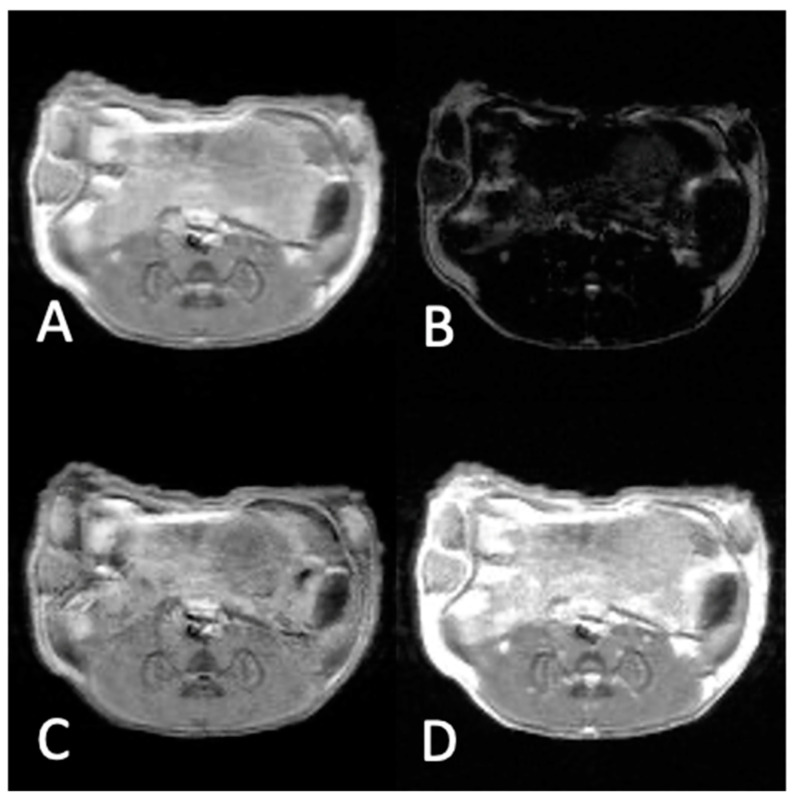
MR images of a mouse bearing TNB tumor pre-injection of the NaDyF_4_/NaGdF_4_ contrast agent: (**A**) T_1_w. (**B**) T_2_w. (**C**) Image obtained by the subtraction of (**B**) from (**A**). (**D**) Image obtained by the addition of (**A**,**B**). The highest contrast is obtained in image (**C**) and is equal to TC = 1.52.

**Figure 8 materials-16-03096-f008:**
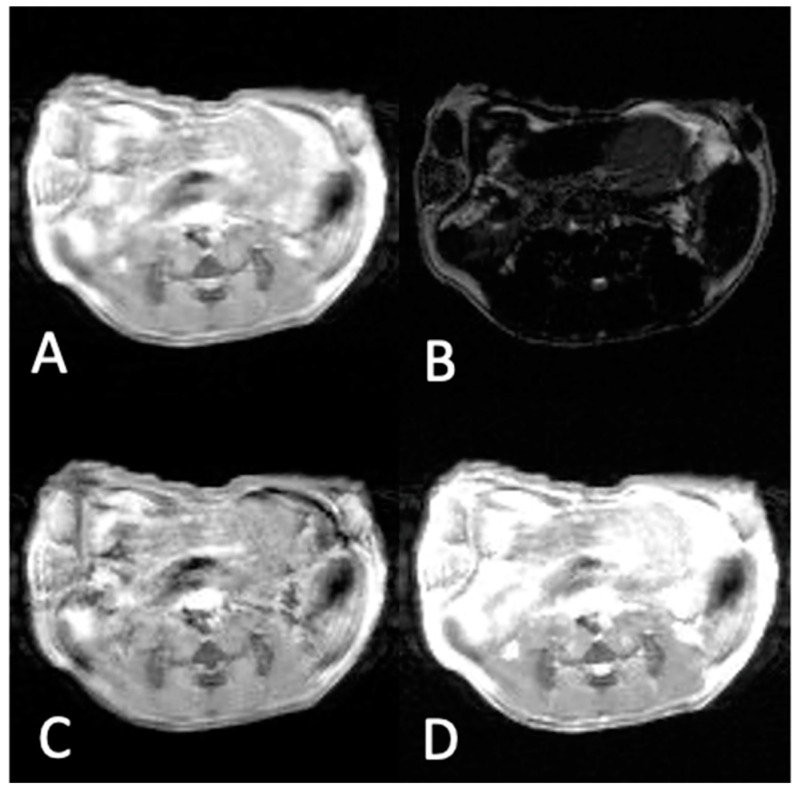
MR images of a mouse bearing TNB tumor 15 min post-injection of non-targeted NaDyF_4_/NaGdF_4_ contrast agent: (**A**) T_1_w. (**B**) T_2_w. (**C**) Image obtained by the subtraction of (**B**) from (**A**). (**D**) image obtained by the addition of (**A**,**B**). The highest tumor contrast is obtained in image (**C**) and is equal to TC = 1.56.

**Table 1 materials-16-03096-t001:** The signal values from the phantom model of tumor (S_tu_), normal adjacent tissue (S_ti_), and the corresponding contrast, were obtained before and after the application of the “perfect” T_1_, T_2_, and T_1_/T_2_ contrast agents. Contrast values (TC) obtained by subtraction (T_1_w − T_2_w) and addition (T_1_w + T_2_w) of signals are also shown.

		S_tu_	S_ti_	TC = S_tu_/S_ti_
Pre-injection	T_1_w	100	100	1.00
	T_2_w	70	70	1.00
	T_1_w − T_2_w	30	30	1.00
	T_1_w + T_2_w	170	170	1.00
Post-T_1_ contrast injection	T_1_w	150	100	1.50
	T_2_w	70	70	1.00
	T_1_w − T_2_w	80	30	2.67
	T_1_w + T_2_w	120	170	0.71
Post-T_2_ contrast injection	T_1_w	100	100	1.00
	T_2_w	35	70	0.47
	T_1_w − T_2_w	65	30	2.17
	T_1_w + T_2_w	222	208	1.07
Post-T_1_/T_2_ contrast injection	T_1_w	150	100	1.50
	T_2_w	35	70	0.50
	T_1_w − T_2_w	115	30	3.83
	T_1_w + T_2_w	185	170	1.09

**Table 2 materials-16-03096-t002:** The values of S_tu_, S_ti_, and TC pre- and 15 min post-injection of non-targeted NaDyF_4_/NaGdF_4_ contrast agent obtained from the images in Figure 7 and Figure 8. The highest tumor contrast is observed for subtracted images both before and after injection.

		S_tu_	S_ti_	TC = S_tu_/S_ti_
Pre-injection	T_1_w	187.0	162.0	1.15
	T_2_w	20.0	60.9	0.33
	T_1_w − T_2_w	167.0	110.0	1.52
	T_1_w + T_2_w	204.0	204.0	1.00
Post T_1_/T_2_ contrast injection	T_1_w	230.0	186.0	1.24
	T_2_w	18.6	57.6	0.32
	T_1_w − T_2_w	216.0	138.0	1.56
	T_1_w + T_2_w	239.0	219.0	1.09

## Data Availability

Not applicable.

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
