# Peer review of "Contrast Enhancement in MRI Using Combined Double Action Contrast Agents and Image Post-Processing in the Breast Cancer Model"

_materials, 2023, doi:10.3390/ma16083096_

Round 1
Reviewer 1 Report
Comments on MDPI Materials
The authors have simulated and evaluated the image contrast change of the tumor phantom model. Based on the simulation results, the authors have performed the in vivo MRI experiments and evaluated the image contrast after administration of NaDyF4/NaGdF4. The highlight of the manuscript seems to be the methods for analyzing the imaging contrast of MRI and showing fewer correlations with the “materials”. Therefore, I suggest the authors revise the manuscript by addressing the following concerns.
1. The title of this manuscript looks like a review instead of a research article. The title conveys limited information to the readers.
2. How did the authors fabricate the NaDyF4/NaGdF4? The synthesis processes and purifications should be given. The characterizations of NaDyF4/NaGdF4 are also lacking. As the journal is titled “materials”, these basic experimental data associated with NaDyF4/NaGdF4 must be provided.
3. Scale bars should be added to the MR images. The tumor should be marked in Figures 7 and 8.
4. When did the in vivo MRI take after the injection of NaDyF4/NaGdF4 via the tail vein?
5. What are the applications of contrast agents that reduce both T1 and T2 relaxation times?
6. What are the r1 and r2 values of NaDyF4/NaGdF4 both in aqueous water and 0.9% saline solution?
7. Some latest Gd-based references should be cited:
1) https://doi.org/10.1016/j.biomaterials.2020.120056
2) https://doi.org/10.1021/acsami.2c12719
Author Response
Response to Reviewer 1 Comments
The authors have simulated and evaluated the image contrast change of the tumor phantom model. Based on the simulation results, the authors have performed the in vivo MRI experiments and evaluated the image contrast after administration of NaDyF4/NaGdF4. The highlight of the manuscript seems to be the methods for analyzing the imaging contrast of MRI and showing fewer correlations with the “materials”. Therefore, I suggest the authors revise the manuscript by addressing the following concerns.
Point 1: The title of this manuscript looks like a review instead of a research article. The title conveys limited information to the readers.
Response 1: We agree. Following this comment, we changed the title to “Contrast Enhancement in MRI using Combined Double Action Contrast Agents and Image Post-processing in the Breast Cancer Model”.
Point 2: How did the authors fabricate the NaDyF4/NaGdF4? The synthesis processes and purifications should be given. The characterizations of NaDyF4/NaGdF4 are also lacking. As the journal is titled “materials”, these basic experimental data associated with NaDyF4/NaGdF4 must be provided
Response 2: The details of the synthesis and characterization of the NPs are described in the paper by Dash et al ACS Appl. Mater. Interfaces 202113(21), 24345–24355. It is cited in our manuscript as a reference [14]. As the synthesis method has been already provided, we believe there is no reason to repeat the entire description. However, to address this valuable comment, we added a brief description of the synthesis. Furthermore, we also included the characterization of the nanoparticles. A new Section 2.3 was added to address these points:
2.3 Synthesis and Characterization of the NaDyF4/NaGdF4 Nanoparticles.
The details of the NaDyF4/NaGdF4 synthesis are described elsewhere [14]. Briefly, before the synthesis of the core/shell NPs, cubic-phase (α) NaGdF4 NPs were synthesized, which acted as the sacrificial NPs for shell formation [27]. The multi-step process of the cubic (α) phase of sacrificial NaGdF4 synthesis started from gadolinium oxide, via gadolinium trifluoroacetate, mixed with various chemicals, heating, washing and cooling cycles. Finally, the NP precipitated, was washed with ethanol and dispersed in hexane.
To synthesize NaDyF4, Dysprosium (III) chloride hexahydrate was stirred in oleic acid and 1-octadecene under vacuum for 45 minutes at 120°C. Further steps included heating, cooling, and stirring solutions of methanol containing sodium hydroxide and ammonium fluoride After removing methanol by heating, the previously synthesized sacrificial (α) NaGdF4 nanoparticles dispersed in 1-octadecene (1 mL) were injected into the solution and stirred for 15 minutes to form a core-shell nanostructure. The nanoparticles were precipitated and washed with ethanol and finally, the nanoparticles were dispersed in hexanes [14].
The synthesized NaDyF4/NaGdF4 NPs were analyzed using the X-ray diffraction method (XRD), showing the standard pattern of the hexagonal phase (β) of the NaDyF4 core. Analysis of the particle size distribution using the transmission electron microscopy (TEM) images showed a NaGdF4 shell thickness of 0.6 ±0.1 nm while the core NaDyF4 NP showed a diameter of 21.2 ±0.1 nm and the core/shell NPs had a diameter of 21.8 ± 0.1nm. The relaxivities r1 and r2 of the NaDyF4/NaGdF4 NPs in deionized water were r1 = (9.0 ± 0.2) × 105 mM-1s-1 and r2 = (147.0 ± 7.5) × 105 mM-1s-1 respectively [14].
Point 3: Scale bars should be added to the MR images. The tumor should be marked in Figures 7 and 8.
Response 3: To address this comment, we have added scale bars and ROIs in Figures 7 and 8.
Point 4: When did the in vivo MRI take after the injection of NaDyF4/NaGdF4 via the tail vein?
Response 4: Thank you for pointing out this omission. The imaging was carried out 15 minutes after injection. To address this point, we added this missing information to the text and in the figures.
Point 5: What are the applications of contrast agents that reduce both T1 and T2 relaxation times?
Response 5: To address this point we have added the following paragraph in the Introduction:
The nanoparticles that reduce both T1 and T2 relaxation times have been used as contrast agents to improve the diagnosis of various diseases. Examples of disease diagnoses impacted include an atherosclerotic plaque in the carotid arteries [11], liver metastases [12], identification of white matter lesions in multiple sclerosis (MS) [13], prostate cancer in animal models [14] or in functional MRI to improve the contrast between brain tissues [15].
Point 6: What are the r1 and r2 values of NaDyF4/NaGdF4 both in aqueous water and 0.9% saline solution?
Response 6: Thank you for this valuable comment. We measured relaxivities in deionized water only. The following was added at the end of the new section 2.3:
The r1 and r2 relaxivities of NaDyF4/NaGdF4 in deionized water were r1 = (9.0 ± 0.2) × 105 mM-1s-1and r2 = (147.0 ± 7.5) × 105 mM-1s-1 respectively. This information was added to the new Section 2.3 as per the comment #2.
Point 7: Some latest Gd-based references should be cited:1)https://doi.org/10.1016/j.biomaterials.2020.120056 2) https://doi.org/10.1021/acsami.2c12719
Response 7: To address this comment, we added the mentioned papers to references as [29] and [30] respectively.

Reviewer 2 Report
The present paper is prepared on a high level. The investigation provided here is quite novel and actual, as it is concerned with investigations of the magnetic resonance imaging process, using contrast agents. On this concern frequently used are Gd- and Fe-based contrast agents, reducing T1 and T2 relaxation times, respectively, providing improved cancer detection. Contrast agents changing both T1/T2 times, based on core/shell nanoparticles, have been introduced in the present work, following the recent developments. Results of investigations presented in the paper could definitely have effective and prospective applications first of all in medicine, for the timely cancer detection and subsequent treatment.
The research presented has been efficiently realized using theoretical calculations of the magnetic resonance signal in a phantom tumor model using T1-weighted, T2-weighted and combined images when T1, T2 and T1/T2 targeted agents are applied. The results from the phantom tumor model are followed by in vivo experiments using core/shell NaDyF4/NaGdF4 nanoparticles as T1/T2 non-targeted contrast agent in the animal model of triple negative breast cancer. The results show that the optimal tumor-to-normal adjacent tissue contrast is provided when T1-weighted and T2-weighted magnetic resonance images are subtracted.
In my opinion, the results reported in the paper can be considered as quite original and well developed. Structure of the paper is quite good. The given tables and figures given in the paper are really needed in order to understand the obtained results more clearly. I have appreciated introduction containing a proper review of the state of the art and previous work in the considered field of knowledge, as well as a rather large number of references.
Moreover, the Conclusions section could be extended with introduction of obtained results. Too little information has been provided there.
After this small correction is realized, the paper definitely could be published.
Author Response
Response to Reviewer 2 Comments
The present paper is prepared on a high level. The investigation provided here is quite novel and actual, as it is concerned with investigations of the magnetic resonance imaging process, using contrast agents. On this concern frequently used are Gd- and Fe-based contrast agents, reducing T1 and T2 relaxation times, respectively, providing improved cancer detection. Contrast agents changing both T1/T2 times, based on core/shell nanoparticles, have been introduced in the present work, following the recent developments. Results of investigations presented in the paper could definitely have effective and prospective applications first of all in medicine, for the timely cancer detection and subsequent treatment.
The research presented has been efficiently realized using theoretical calculations of the magnetic resonance signal in a phantom tumor model using T1-weighted, T2-weighted and combined images when T1, T2 and T1/T2 targeted agents are applied. The results from the phantom tumor model are followed by in vivo experiments using core/shell NaDyF4/NaGdF4 nanoparticles as T1/T2 non-targeted contrast agent in the animal model of triple negative breast cancer. The results show that the optimal tumor-to-normal adjacent tissue contrast is provided when T1-weighted and T2-weighted magnetic resonance images are subtracted.
In my opinion, the results reported in the paper can be considered as quite original and well developed. Structure of the paper is quite good. The given tables and figures given in the paper are really needed in order to understand the obtained results more clearly. I have appreciated introduction containing a proper review of the state of the art and previous work in the considered field of knowledge, as well as a rather large number of references.
Thank you very much for the positive comments.
Point 1: The Conclusions section could be extended with introduction of obtained results. Too little information has been provided there. After this small correction is realized, the paper definitely could be published.
Response 1:
To address this point (also indicated by reviewer #3), we have added the following to the discussion:
The result for the phantom tumor model shows that the application of the T1/T2 contrast agent alone improves tumor contrast by 50% (from 1.0 to 1.5) on T1-weighted images and reduced contrast by 50% (from 1.0 to 0.5) on T2-w images. The image subtraction additionally improves contrast over 2-fold (from 1.50 to 3.83) when compared to T1-weighted. For the in vivo experiments, the corresponding values increase by 8% (from 1.15 to 1.24) in T1-weighted images, decrease by 1% (from 0.33 to 0.32) in T2-weighted images and additional improvement due to image subtraction by 12% (from 1.24 to 1.56). Overall, the results show that the image contrast enhancement caused by the T1/T2 contrast agents can be additionally improved by the application of the image subtraction.

Reviewer 3 Report
MANUSCRIPT – materials-2301287
In this article,
Image Processing in Contrast Enhanced MRI
The authors have studied some of the neglected facts when using the MRI contrast agents using the theoretical model, and based on that, an in vivo study has been carried out using the TNBC model. The study's outcome shows an optimal contrast between healthy and cancer regions using T1 & T2-weighted MR images.
Significant comments that should be addressed
Major language checking is necessary; it is better to check with the native speaker.
Title
The title looks pretty general; please modify it based on the study.
Abstract
The abstract is written well; I suggest adding some numerical data to make it more attractive to the readers.
Introduction
Line 34-Before using the acronym of diagnostic modalities, it is better to mention it entirely for the first time and then use the abbreviation.
Overall, the introduction paragraph is crisp and precise; I like how it is explained.
Line 66: I recommend including this reference https://doi.org/10.1016/j.ijpharm.2017.03.071 for core-shell nanoparticles and their use in MRI.
Materials and methods
General comment: Mention the purity/percentage of chemicals, brand/company etc., used for the whole study; it is missing.
Section 2.1
On what basis are the parameters set for the theoretical study, and whether the methodology involves any particular software? Please mention it
Section 2.2
Before explaining the process of inoculating cancer in the mouse model, it is necessary to mention the mouse's physical characteristics, like weight and the details of ethical committee approval.
Section 2.3
There are no specific details on nanoparticles, like whether they are synthesized or commercial. Please include the details for reproducibility
Explain why the nanoparticles sizes of core 20 & shell 0.5 nm were chosen for the study.
Why does this combination of core-shell opt for the study?
Results
Section 3.1
Before giving the experiment details, provide how the theoretical study was made.
Please provide basic characterization studies of nanoparticles like optical, morphological, particle size, etc
Mention why the study has been limited to only one type of nanoparticle size or on what basis it was determined.
Discussion & Conclusion
This section should be improved; it is vague and lacks scientific interpretation.
Author Response
Response to Reviewer 3 Comments
The authors have studied some of the neglected facts when using the MRI contrast agents using the theoretical model, and based on that, an in vivo study has been carried out using the TNBC model. The study's outcome shows an optimal contrast between healthy and cancer regions using T1 & T2-weighted MR images.
Point 1: Major language checking is necessary; it is better to check with the native speaker.
Response 1: To address this comment the manuscript was reviewed and corrected by two native speakers. However, should the reviewer be not satisfied with the corrections, we would appreciate providing examples of incorrect grammar. Alternatively, we could also request a review by an external editor.
Point 2: The title looks pretty general; please modify it based on the study.
Response 2: To address this point (also indicated by reviewer #1) we changed the title to “Contrast Enhancement in MRI using Combined Double Action Contrast Agents and Image Post-processing in the Breast Cancer Model”.
Point 3: The abstract is written well; I suggest adding some numerical data to make it more attractive to the readers.
Response 3: To address this comment, we added the sentence below. We could not expand on the topic more due to the 200 words limit. To keep the 200-word limit and to include numerical data, we removed a few words as shown in the now-corrected Abstract. However, we extended the discussion at the end of the manuscript to better address this point.
The results show that the subtraction of T2-weighted from T1-weighted MR images provides an additional increase of the tumor contrast: over 2-fold in the phantom model and 12% in the in vivo experiment.
Point 4: Line 34-Before using the acronym of diagnostic modalities, it is better to mention it entirely for the first time and then use the abbreviation.
Response 4: Thank you for pointing out our omission. We added the appropriate explanation:
Magnetic resonance imaging (MRI) provides the best soft tissue contrast among diagnostic imaging modalities such as computed tomography (CT), positron emission tomography (PET) or X-ray.
Point 5: Line 66: I recommend including this reference https://doi.org/10.1016/j.ijpharm.2017.03.071 for core-shell nanoparticles and their use in MRI.
Response 5: To address this comment, we added this reference (line 60)(Ravichandran M, Oza G, Velumani S, Ramirez JT, Vera A, Leija L, Design and evaluation of surface functionalized superparamagneto-plasmonic nanoparticles for cancer therapeutics, Int J Pharmaceutics, 2017, 524(1-2): 16-29). It is cited in our manuscript as [8].
Point 6: General comment: Mention the purity/percentage of chemicals, brand/company etc., used for the whole study; it is missing.
Response 6: The reviewer #1 had a similar comment. The details of the synthesis and characterization of the NPs are described in the paper by Dash et al ACS Appl. Mater. Interfaces 202113(21), 24345–24355. It is cited in our manuscript as a reference [14]. As the synthesis method has been already provided, we believe there is no reason to repeat the entire description. However, to address this valuable comment, we added a brief description of the synthesis. Furthermore, we also included the characterization of the nanoparticles. A new Section 2.3 was added to address these points:
2.3 Synthesis and Characterization of the NaDyF4/NaGdF4 Nanoparticles.
The details of the NaDyF4/NaGdF4 synthesis are described elsewhere [14]. Briefly, before the synthesis of the core/shell NPs, cubic-phase (α) NaGdF4 NPs were synthesized, which acted as the sacrificial NPs for shell formation [27]. The multi-step process of the cubic (α) phase of sacrificial NaGdF4 synthesis started from gadolinium oxide, via gadolinium trifluoroacetate, mixed with various chemicals, heating, washing and cooling cycles. Finally, the NP precipitated, was washed with ethanol and dispersed in hexane.
To synthesize NaDyF4, Dysprosium (III) chloride hexahydrate was stirred in oleic acid and 1-octadecene under vacuum for 45 minutes at 120°C. Further steps included heating, cooling and stirring solutions of methanol containing sodium hydroxide and ammonium fluoride After removing methanol by heating, the previously synthesized sacrificial (α) NaGdF4 nanoparticles dispersed in 1-octadecene (1 mL) were injected into the solution and stirred for 15 minutes to form a core-shell nanostructure. The nanoparticles were precipitated and washed with ethanol and finally, the nanoparticles were dispersed in hexanes [14].
The synthesized NaDyF4/NaGdF4 NPs were analyzed using the X-ray diffraction method (XRD), showing the standard pattern of the hexagonal phase (β) of the NaDyF4 core. Analysis of the particle size distribution using the transmission electron microscopy (TEM) images showed a NaGdF4 shell thickness of 0.6 ±0.1 nm while the core NaDyF4 NP showed a diameter of 21.2 ±0.1 nm and the core/shell NPs had a diameter of 21.8 ± 0.1nm. The relaxivities r1 and r2 of the NaDyF4/NaGdF4 NPs in deionized water were r1 = (9.0 ± 0.2) × 105 mM-1s-1 and r2 = (147.0 ± 7.5) × 105 mM-1s-1 respectively [14].
Point 7: On what basis are the parameters set for the theoretical study, and whether the methodology involves any particular software? Please mention it
Response 7: We did not use any software in the tumor model section.
The assumed changes in signals due to contrast agent injection are based on the reports on the application of breast contrast agents showing 50-150% enhancement in breast MRI after contrast agent injection (e.g. Hendrick ER, High-Quality Breast MRI, Radiol Clin N Am 2014, 52:547–562). The intensity units (100 and 70) were arbitrarily selected.
To clarify this point in more detail:
The provided arbitrary values of pixel intensities and their changes come from the action of MRI contrast agents: the T1 contrast agents reduce T1 values hence increasing signal intensity in T1w images (so-called “hyperintense image”), while T2 contrast agents reduce T2 values and decrease signal intensity in T2w images. This causes a reduction of contrast (“hypointense images”) when T2 contrast and T2-w pulse sequence are used. Some authors refer to this effect as a “signal nulling”. (For more please see for example: Sunil N. et al. MR Contrast Agents for Liver Imaging: What, When, How, RadioGraphics 2006 26:6, 1621-1636). In our model, we assumed that the T1 contrast agent changes T1 only while T2 contrast changes T2 only values. We called it a “perfect contrast” because, in reality, the T1 contrast agent changes mostly T1 and, to some degree, T2 value (~10% depending on the field strength and contrast used). Similarly – T2 contrast changes mostly T2 and to some degree T1 (also by about 10%). In the theoretical model we presented the “perfect” T1 and T2 contrast agent that changes only T1 or T2 respectively. The “perfect” T1/T2 contrast agent was assumed to change both T1 and T2 by a similar value hence signal intensity (by ~50%).
The action of a real contrast agent in vivo is more complex due to different accumulations of the contrast agents in the tumor cells and in the surrounding tissue. This behavior depends on the type of tumor and the contrast agent. For example, targeted contrast agents (namely synthesized with, for example, an antibody specific to the tumor cells) accumulate in tumors in larger quantities than non-targeted.
As explained above the impact of a contrast agent on the tumor MRI contrast is a complex issue and, as we believe, outside of the scope of this paper. However, to address this valid point we referred an interested reader (by adding 2 references) to two review papers on contrast agents (Strijkers GJ, Mulder WJ, van Tilborg GA, Nicolay K. MRI contrast agents: current status and future perspectives. Anticancer Agents Med Chem. 2007, 7(3):291-305. and Xiao Y, Paudel R, Liu J, Ma C, Zhang Z, Zhou S. MRI contrast agents: Classification and application (Review). Int J Mol Med 2016, 38:1319-1326) cited in the manuscript as [25,26].
Furthermore, the applied MRI pulse sequence may completely change image intensity, for example by a simple application of an inversion rf pulse before the imaging sequence would provide different imaging contrast that depends not only on T1, T2 but also on the inversion recovery time.
In Summary: we believe, our model provides a good estimation of the behaviour of contrast agents in vivo in a general case hence can be applied to other tumors.
To briefly address the above complex points, we modified the following paragraph :
Based on previous reports on cancer MRI and, in particular, on MRI of breast cancer studies (e.g., Hendrick ER, High-Quality Breast MRI, Radiol Clin N Am 2014, 52:547–562, cited in the manuscript as [24]) showing about 50% TC enhancement caused by contrast agent injection, we considered a “perfect” tumor phantom model. This allowed us to study TC without the influence of factors occurring in vivo (e.g., variable contrast absorption, tumor metabolism, tumor stage, contrast agent concentration etc.) and MRI pulse sequence type and its parameters. We first assumed that tumor and surrounding tissue signals are identical in both T1w (equal to 100 arbitrary units (au)) and in T2w images (equal to 70 au). We also assumed “perfect” targeted contrast agents, accumulating in the tumor only. We further assumed, for the T1 contrast agent, the signal increases by 50% in the tumor only and does not change in the normal adjacent tissue. Similarly, for T2 contrast agent: the signal from the tumor decreases by 50% and is unchanged in normal adjacent tissue in T2w MR images. For T1/T2 contrast agent, we assumed a 50% signal increase and a 50% reduction in T1w and T2w images respectively. In addition to providing signal values obtained from T1w and T2w images, we also added and subtracted images (Table 1).
Point 8: Before explaining the process of inoculating cancer in the mouse model, it is necessary to mention the mouse's physical characteristics, like weight and the details of ethical committee approval.
Response 8: To address this point, we added the weight of the mouse 35.6g, heart rate 310-840 beats/min, and body temp 36.5-38.0 .
The approval of the Animal Care Committee was mentioned at the end of the text in the Institutional Review Board Statement: “Animal experiment approved by the local Animal Care Committee”. However, to address this important comment, we also added the following sentence in the text (line 214): The animal experiment was approved by the local Animal Care Committee (number AC13-0202).
Point 9: There are no specific details on nanoparticles, like whether they are synthesized or commercial. Please include the details for reproducibility.
Response 9: The same point was brought forward by the reviewer #1. The details of the synthesis and characterization of the NPs are described in the paper by Dash et al ACS Appl. Mater. Interfaces 202113(21), 24345–24355. It is cited in our manuscript as a reference [14]. As the synthesis method has been already provided, we believe there is no reason to repeat the entire description. However, to address this valuable comment, we added a brief description of the synthesis. Furthermore, we also included the characterization of the nanoparticles. A new Section 2.3 was added to address these points:
2.3 Synthesis and Characterization of the NaDyF4/NaGdF4 Nanoparticles.
The details of the NaDyF4/NaGdF4 synthesis are described elsewhere [14]. Briefly, before the synthesis of the core/shell NPs, cubic-phase (α) NaGdF4 NPs were synthesized, which acted as the sacrificial NPs for shell formation [27]. The multi-step process of the cubic (α) phase of sacrificial NaGdF4 synthesis started from gadolinium oxide, via gadolinium trifluoroacetate, mixed with various chemicals, heating, washing and cooling cycles. Finally, the NP precipitated, was washed with ethanol and dispersed in hexane.
To synthesize NaGdF4, Dysprosium (III) chloride hexahydrate was stirred in oleic acid and 1-octadecene under vacuum for 45 minutes at 120°C. Further steps included heating, cooling and stirring solutions of methanol containing sodium hydroxide and ammonium fluoride After removing methanol by heating, the previously synthesized sacrificial (α) NaGdF4 nanoparticles dispersed in 1-octadecene (1 mL) were injected into the solution and stirred for 15 minutes to form a core-shell nanostructure. The nanoparticles were precipitated and washed with ethanol and finally, the nanoparticles were dispersed in hexanes [14].
The synthesized NaDyF4/NaGdF4 NPs were analyzed using the X-ray diffraction method (XRD), showing the standard pattern of the hexagonal phase (β) of the NaDyF4 core. Analysis of the particle size distribution using the transmission electron microscopy (TEM) images showed a NaGdF4 shell thickness of 0.6 ±0.1 nm while the core NaDyF4 NP showed a diameter of 21.2 ±0.1 nm and the core/shell NPs had a diameter of 21.8 ± 0.1nm. The relaxivities r1 and r2 of the NaDyF4/NaGdF4 NPs in deionized water were r1 = (9.0 ± 0.2) × 105 mM-1s-1 and r2 = (147.0 ± 7.5) × 105 mM-1s-1 respectively [14].
Point 10: Explain why the nanoparticles sizes of core 20 & shell 0.5 nm were chosen for the study.
Response 10: These were the only NPs provided by our collaborators. Please see also point 13.
Point 11: Before giving the experiment details, provide how the theoretical study was made.
Response 11: We believe we addressed this comment in point 7. However, should the reviewer find the details to be insufficient we would appreciate an explanation of what the reviewer exactly meant by “how the theoretical study was made”.
Point 12: Please provide basic characterization studies of nanoparticles like optical, morphological, particle size, etc.
Response 12: This comment was addressed in point 9.
Point 13: Mention why the study has been limited to only one type of nanoparticle size or on what basis it was determined.
Response 13: This comment is similar to the point 10. As we explained in point 10 these NPs were the only NPs provided by our collaborators. However, we see no reason why using other NPs would provide other results. In contrary, as mentioned in the Discussion section, NPs with higher r1 and r2 would provide even more evident increase in TC. The increase in r1 and r2 can be achieved by increasing both the shell and core sizes. In this particular case, the increase in the Gd shell size would increase r1 hence providing the desired improvement in TC. This statement is further supported by the results obtained in our “perfect” phantom model, that assumed no specific size of NPs.
To address this point, we added to the Discussion as follows:
In the presented studies only one type of NP was used to show the general concept of image analysis. The NaDyF4/NaGdF4 NPs used in the in vivo study had a core/shell size of 20nm/0.5nm. However, we expect the observed increase in TC would be achieved independently of the size of the NP. NPs with higher r1 and r2 would provide an even more evident increase in TC. The increase in r1 and r2 can be achieved by increasing both the shell and core sizes. For the NaDyF4/NaGdF4 NP, an increase in the Gd shell size would increase r1 hence providing the desired improvement in TC. The independence of the TC enhancement due to the applied image post-processing is also supported by the theoretical analysis of the phantom model, where an arbitrary contrast was considered. However, further studies with other NPs are required to find their optimum parameters (e.g., core/shell ratio and size) to obtain maximum TC.
Point 14: This section should be improved; it is vague and lacks scientific interpretation.
Response 14: To address this point, we have added numerical values to the Discussion and moved the last paragraph from the Discussion to the Conclusion.
The result for the phantom tumor model shows that the application of the T1/T2 contrast agent alone improves tumor contrast by 50% (from 1.0 to 1.5) on T1-weighted images and reduced contrast by 50% (from 1.0 to 0.5) on T2-w images. The image subtraction additionally improves contrast over 2-fold (from 1.50 to 3.83) when compared to T1-weighted. For the in vivo experiments, the corresponding values increase by 8% (from 1.15 to 1.24) in T1-weighted images, decrease by 1% (from 0.33 to 0.32) in T2-weighted images and additional improvement due to image subtraction by 12% (from 1.24 to 1.56). Overall, the results show that the image contrast enhancement caused by the T1/T2 contrast agents can be additionally improved by the application of the image subtraction. Furthermore, TC is highest for the T1/T2 contrast agent. Subsequently, potential drawbacks associated with lower contrast provided by the T1/T2 contrast agents can be rectified.

Round 2
Reviewer 1 Report
This work can be published.
Reviewer 3 Report
The authors have made the substantial modification, it is now can be accepted in the present form.